# MRI-Based Phenotyping for Osteosarcopenic Adiposity in Subjects from a Population-Based Cohort

**DOI:** 10.3390/geriatrics9060150

**Published:** 2024-11-14

**Authors:** Elke Maurer, Susanne Rospleszcz, Wolfgang Rathmann, Barbara Thorand, Annette Peters, Christopher L. Schlett, Fabian Bamberg, Lena Sophie Kiefer

**Affiliations:** 1Department for Trauma and Reconstructive Surgery, BG Unfallklinik Tübingen, 72076 Tübingen, Germany; 2Institute of Epidemiology, Helmholtz Zentrum München, German Research Center for Environmental Health, 85764 Neuherberg, Germany; susanne.rospleszcz@helmholtz-muenchen.de (S.R.); thorand@helmholtz-muenchen.de (B.T.); peters@helmholtz-muenchen.de (A.P.); 3Medical Faculty, Institute for Medical Information Processing, Biometry and Epidemiology, Ludwig-Maximilians-Universität München, 80539 Munich, Germany; 4German Center for Diabetes Research (DZD), Partner Site Neuherberg, 85764 Neuherberg, Germany; wolfgang.rathmann@ddz.uni-duesseldorf.de; 5Department of Diagnostic and Interventional Radiology, Medical Center–University of Freiburg, 79106 Freiburg, Germany; christopher.schlett@uniklinik-freiburg.de (C.L.S.); fabian.bamberg@uniklinik-freiburg.de (F.B.); 6Department of Nuclear Medicine and Clinical Molecular Imaging Otfried-Müller-Straße 14, 72076 Tübingen, Germany

**Keywords:** osteosarcopenic adiposity, magnetic resonance imaging, phenotyping, imaging biomarker, population-based cohort imaging

## Abstract

**STUDY IMPORTANCE QUESTIONS:**

**What is already known about this subject?**

**What are the new findings in your manuscript?**

**How might your results change the direction of research or the focus of clinical practice?**

**Abstract:**

**Objective:** Imaging biomarkers of bone, muscle, and fat by magnetic resonance imaging (MRI) may depict osteopenia, sarcopenia, and adiposity as the three different conditions of osteosarcopenic adiposity (OSA). **Methods:** Subjects from a prospective, population-based case–control study underwent a health assessment and 3 Tesla whole-body MRI scan. Imaging biomarkers of bone (bone marrow fat-fraction (BMFF)), skeletal muscle (skeletal muscle FF (SMFF)), and fat (total adipose tissue (TAT)) were determined. Participants were allocated to one phenotype according to the OSA complex. **Results:** Among 363 participants forming the study cohort, 81 (22.3%, 48.1% males, 62.4 ± 6.9 years) were allocated into the OSA subgroup. Participants with an OSA phenotype were significantly older compared to all remaining subjects and showed the highest grades of SMFF (all *p* < 0.005). Together with subjects from the osteopenic sarcopenia group, OSA subjects exhibited the highest amounts of BMFF and together with the three other adiposity-containing subgroups also exhibited the highest BMIs. The highest prevalence of an impaired glucose tolerance as well as significantly higher blood pressure, blood dyslipidemia, and hepatic steatosis was found in the OSA subgroup (all *p* < 0.005). **Conclusions:** MR biomarkers of bone, skeletal muscle and fat are feasible for body composition phenotyping and may allow for targeted risk stratification in suspected OSA syndrome.

## 1. Introduction

Osteosarcopenic adiposity (OSA) is characterized as the coexistence of bone and muscle deterioration in conjunction with excess body fat, resulting in impaired functionality and systemic metabolic dysregulation [1,2]. The three individual conditions of OSA are defined as follows: (1) a decrease in bone mineral density (osteopenia/osteoporosis), (2) a decrease in muscle mass, strength, and/or functional capacity (sarcopenia), and (3) an increase in adipose tissue and ectopic lipid deposition (adiposity, fat mal- and redistribution into visceral compartments, bone, and muscle tissue) [1,2,3]. However, these three individual conditions do not only coincide to different degrees as a comorbidity (“hazardous triad”). In fact, since they share common risk factors such as age, gender, and physical inactivity, and are closely linked in nature, OSA may codevelop, beginning with either one condition and progress into the full triad if left untreated [2,4]. Major complications are not only frailty and physical disability with an increased risk for falling, but also an aggravation of the cardio-metabolic profile with consecutively exacerbated negative health outcomes and increased all-cause mortality [5,6,7].

OSA syndrome features a very complex and multifactorial pathophysiology (e.g., genetic, endocrine, and environmental factors), which is still not sufficiently understood for the most part. In this context, fat redeposition in visceral, osseous, and muscular compartments has been described as a potential contributor, causing a lipotoxic microenvironment to the surrounding tissues with altered physiological processes and ultimately leading to local and systemic dysregulations [8,9].

Apart from adiposity, which is phenotypically the most apparent condition, and secondary clinical manifestations and/or complications of OSA, osteopenia and sarcopenia itself often remain underdetected and undertreated. Given the high rate of their comorbidity and high risk of complications, the clinical suspicion or diagnosis of each component of the OSA triad should prompt targeted examination for the other ones [3].

Different modalities are available to characterize and quantify bone, muscle and fat. The most commonly used techniques are dual-energy X-ray absorptiometry (DEXA), bioelectrical-impedance analysis (BIA), and magnetic resonance imaging (MRI) [9,10]. In contrast to DEXA and BIA, MRI allows for the simultaneous characterization and quantification of skeletal muscle as well as for a profound analyzation of adipose tissue compartments (e.g., visceral adipose tissue (VAT) and subcutaneous adipose tissue (SAT)) and ectopic lipid deposits. But besides muscle and fat, bone assessment (e.g., fat infiltration) is also possible by MRI [11], thus enabling a sophisticated analysis of body composition phenotypes specifically regarding OSA and its components [9,12].

In this exploratory analysis, we systematically performed body composition analysis by determining MR imaging biomarkers of bone, muscle, and adipose tissue in subjects from a population-based cohort. Based on these determinants, subjects were allocated to one phenotype according to the OSA complex (median divided and sex-specific). Characteristics of these different MRI-phenotypes were compared and correlations to cardio-metabolic risk factors were analyzed. Our hypothesis was that body composition phenotyping by MRI may be feasible and allows for risk stratification for OSA syndrome.

## 2. Methods

### 2.1. Study Design

Subjects were derived from the KORA-FF4 sub-study (06/2013-09/2014, *n* = 2279), the second follow-up study of a cross-sectional case–control study nested in a prospective cohort from the Cooperative Health Research in the Region of Augsburg (KORA) in Germany. The study was approved by the ethics committee of the Bavarian Chamber of Physicians, Munich, Germany, and the local institutional review board of the Ludwig-Maximilians-University Munich, Germany. Participants provided written informed consent [13].

### 2.2. MR Imaging Protocol and Biomarkers of Osteosarcopenic Adiposity

In total, 400 eligible subjects from the KORA FF4-cohort underwent whole-body MRI according to previously described inclusion/exclusion criteria. All MRI examinations were performed within 3 months after the clinical examination at the study center. MRI examinations were performed in supine position on a 3 Tesla Magnetom Skyra (Siemens Healthineers, Erlangen, Germany) using an 18-channel body surface coil and a table-mounted spine matrix coil. The imaging protocol and technical specificities have been described previously [13].

#### 2.2.1. Bone Marrow Fat Fraction—Osteopenia

Osteopenia and/or osteoporosis, defined as reduced bone mineral density (BMD), has previously been described as the obesity of bone [14,15]. Referring to this, although not conclusively proven to date and still subject to critical discussion, recent data suggested that bone marrow fat fraction (BMFF), reflecting adipocytes, correlated inversely with BMD [16]. One explanation for this reciprocal relationship may be that bone resorption cavities consecutively refill with yellow bone marrow (=fat marrow) [15,17], this being reflected in the fact that with progressively decreasing BMD through aging, bone marrow fat steadily increases. Therefore, BMFF may be used as an imaging biomarker for an osteopenic phenotype.

Coronally acquired 2-point Dixon T1-weighted VIBE CAIPIRINHA sequences (time to repetition (TR): 4.06 ms; time to echo (TEs): 1.26 ms and 2.49 ms; flip angle 4°; slice thickness 1.7 mm) were used to determine BMFF in L1 and L2 vertebrae. Water and fat selective images were automatically calculated by the manufacturer’s software. The segmentation of bone marrow has been described before [11]. In brief, regions of interest (ROIs) quantifying BMFF were manually drawn in one coronal image in the middle of the anterior–posterior diameter of the vertebral body (Figure 1.1). Thus, the segmented ROIs included only cancellous while excluding cortical bone. BMFF values were then calculated as the mean value (fat image) divided by mean value (fat + water image) [11].

#### 2.2.2. Skeletal Muscle Fat Fraction and Muscle Mass—Sarcopenia

Analogous to bone, sarcopenia has been described as the obesity of muscle by some authors. In addition to the mere loss of muscle mass, fat redistribution with ectopic infiltration in muscle tissue (myosteatosis) has recently been delineated as an important determinant of frailty in aging, and as one key component of sarcopenia [18]. Furthermore, the decline in muscle mass and strength/functionality occurring with, e.g., progressive aging is (comparable to bone) related with increased myosteatosis. In this context, the single level-based quantification of skeletal muscle fat fraction (SMFF) and cross-sectional area (CSA) determining myosteatosis and muscle mass has been shown to be representative for the entire body [19]. In this study, SMFF has been used as an imaging biomarker in order to describe a sarcopenic phenotype.

Muscle segmentation was performed at level L3 vertebra on axial slices on T2*-corrected, multi-echo 3D-gradient-echo Dixon-based sequences (multi-echo Dixon: (TR: 8.90 ms; TEs: 1.23 ms, 2.46 ms, 3.69 ms 4.92 ms, 6.15 ms, and 7.38 ms; flip angle 4°, slice thickness 4 mm)). Details of the segmentation procedure and post-processing algorithm have been described recently [20]. In brief, each muscle compartment (right and left psoas major and autochthonous back muscles (=erector spinae muscles)) was manually segmented according to standardized, anatomical landmarks (Figure 1.2). The degree of myosteatosis was determined as the mean SMFF and muscle mass as the CSA.

#### 2.2.3. Total, Visceral, and Subcutaneous Adipose Tissue—Adiposity

Adiposity is characterized by an excess of adipose tissue and increase in ectopic lipid deposition with fat mal- and redistribution not only into visceral and subcutaneous compartments, but also into organs (e.g., bone, muscle, and liver). Measurements of adipose compartments by MRI quantifying the total (TAT), visceral (VAT), and subcutaneous adipose tissue (SAT) have been established and may therefore be feasible imaging biomarkers to characterize an adipose phenotype [13,21].

Trunk adipose tissue compartments were segmented and quantified by a semi-automated algorithm based on fuzzy-clustering. Therefore, a fat-selective tomogram (slice thickness 5 mm) was calculated based on a three-dimensional in/opposed-phase VIBE-Dixon sequence (dual-echo Dixon) (TR: 4.06 ms; TEs: 1.26 ms, 2.49 ms; flip angle 9°; slice thickness: 1.7 mm). Volumes of SAT and VAT were quantified from the cardiac apex to the femoral head and from the diaphragm to the femoral head, respectively. TAT was calculated as the sum of VAT + SAT (Figure 1.3) [13,22].

#### 2.2.4. Phenotypic Subgroups of the OSA Complex

Imaging biomarkers of bone (BMFF), muscle (SMFF), and adipose tissue (TAT) were used to allocate subjects to one phenotype according to the OSA complex (Table 1, Figure 2). An example of two study participants with higher and lower TAT, SMFF, and BMFF is provided in Figure 3. In this study, subjects with a mean BMFF_L1&L2_ greater than the sex-specific median were assigned with any osteopenic phenotype. Likewise, subjects with a TAT greater than the sex-specific median were allocated to an adipose phenotype. The sarcopenic phenotype was defined as an SMFF_psoas&authochtonous_ greater than the sex-specific median. Intersections based on these definitions were built. Subjects which were equal or below the sex-specific median in all three categories were classified with a “*normal*”/healthy phenotype. 

### 2.3. Health Assessment and Covariates

All participants underwent a comprehensive health assessment with standardized interviews and physical examinations in order to determine the main characteristics and demographics (e.g., age and gender) as well as other, important covariates (e.g., cardio-metabolic risk factors).

#### 2.3.1. Anthropometry and BIA

The body mass index (BMI) was calculated as body weight in kg divided by body height squared in m^2^. Waist circumference was measured at the smallest abdominal circumference or, in obese subjects, in the midpoint of the lowest rib and the upper margin of the iliac crest.

Whole-body BIA scans were acquired using a body impedance analyzer (BIA 2000-S, Data-Input, Pöcking, Germany) with an operating frequency of 50 kHz at 0.8 mA. Ohmic resistance was measured between the dominant hand wrist and dorsum and the dominant foot angle and dorsum in supine position. Total body fat content (in %) as well as lean body and appendicular muscle mass indices (in kg, normalized to subjects’ body height squared in m^2^) were collected.

#### 2.3.2. Glucose Tolerance, Lipid Metabolism, Vitamin D

A 75 g oral glucose tolerance test (OGTT) was performed for all participants who had not yet been diagnosed with diabetes mellitus (DM). According to the WHO definition, subjects were classified with an impaired glucose tolerance, either with established type 2 DM (T2DM) or prediabetes (two-hour plasma glucose following a 75 g OGTT ≥ 7.8 mmol/L and/or fasting plasma glucose (FPG) ≥ 5.6 mmol/L) and as normoglycemic (OGTT < 7.8 mmol/L and/or FPG < 5.6 mmol/L) [23].

Blood samples for the laboratory test were collected and blood levels of fasting glucose, HbA1c, blood lipids (HDL, LDL, and triglycerides) and vitamin D/calciferol were assessed.

#### 2.3.3. Physical Activity

Participants’ physical activity was recorded by a single-choice five-level-each multiple-choice question. Subjects were categorized as physically active (regular physical activity ≥ 1 h/week) or as physically inactive (irregular physical activity < 1 h/week, almost no/no physical activity) [24].

#### 2.3.4. Other Covariates

Blood pressure was measured using a validated automatic device (OMRON HEM 705-CP). Thereby, systolic and diastolic blood pressures were measured three times at the right arm in a sitting position after at least 5 min of rest and with a pause of at least 3 min between the three readings. For this analysis, the calculated mean of the second and third measurements was used [25]. Osteoarthritis of the hip joint was categorized by MRI according to the Kellgren–Lawrence classification using axial dual-echo Dixon and coronal T2w single-shot fast-spin echo (SS-FSE/HASTE) sequences (dual-echo Dixon: sequence parameters as described above; T2 HASTE: TE 91 ms, TR 1000 ms, flip angle: 131°, partition segments: 5 mm). Thereby, joint gap narrowing, osteophytic lipping, and subchondral changes (e.g., sclerosis, pseudocysts) were analyzed [26]. Intervertebral disk degeneration was categorized by MRI according to the Pfirrmann grading system into grade 1 to 5 on T2-weighted single-shot fast spin-echo sequences (TR: 1000 ms, TE: 91 ms, flip angle 131°, slice thickness 5 mm). Thereby, structure, the distinction between the nucleus and annulus, signal intensity, and the height of the intervertebral disk were analyzed for each segment from lumbar vertebrae 1 to 5. Hepatic steatosis was assessed based on intrahepatic lipid content using a T2*-corrected, multi-echo Dixon sequence (parameters as described above) with ROIs being placed in the right and left liver lobes (segments 8 and 2). Hepatic fat fraction (HFF) was calculated as the average of the right and left lobe measurements. Fatty liver disease (FLD) was defined according to the Clinical Practice Guidelines for the management of non-alcoholic fatty liver disease as an HFF > 5.6% [27]. Somatic pain symptoms were evaluated by self-reports of any pain symptoms in the head and back, joints, arms and/or legs.

### 2.4. Statistical Analysis

The characteristics of the study sample are presented as the mean and standard deviation or median with interquartile range (IQR) for continuous data, and counts and percentages for categorical data. Differences according to the phenotypic subgroups of the OSA complex, as outlined above, were graphically evaluated by boxplots. One-way ANOVA was used to assess whether continuous covariates differed in their mean value across phenotypic subgroups and an *χ*^2^-Test was used to assess whether categorical covariates differed in their distribution across phenotypic subgroups. In this exploratory analysis, *p*-values < 0.05 were considered to denote statistical significance. Statistical analysis was performed using R v4.1.2.

## 3. Results

Among 400 subjects who underwent whole-body MRI, 37 subjects (12.2%) were excluded due to insufficient image quality or incomplete MRI data sets of one of the sequences included. Thus, the study cohort consisted of *n* = 363 subjects (mean age: 56.0 ± 9.1 years, 57.6% male sex, mean BMI: 27.9 ± 4.6 kg/m^2^). Demographics and body composition characteristics are provided in Table 2 and Table 3.

In total, 81 subjects (22.3%) were classified with an OSA phenotype and 88 subjects (24.2%) with a “*normal*” phenotype, lacking any trait of the OSA complex. Further, 194 subjects (53.5%) demonstrated at least one or two phenotypic components of the OSA complex and were allocated to one subgroup accordingly (Figure 2). Subjects with an OSA phenotype were significantly older compared to subjects with a “*normal*” phenotype (62.4 ± 6.9 years vs. 49.0 ± 7.3 years; *p* < 0.005).

### 3.1. Adipose Phenotype

In total, 179 subjects (49.3%, 58.7% male gender) were allocated to any subgroup containing the adipose phenotype (isolated or comorbid) (Figure 1 and Figure 4). In comparison, n = 262 subjects from this cohort would have been defined as overweight (n = 156, 43.0%) or obese (n = 106, 29.2%) based on their BMI according to the well-established WHO definitions (BMI 25–30 kg/m^2^ and ≥30 kg/m^2^, respectively) [28]. Thereby, only six subjects (6% of all participants with a BMI < 25 kg/m^2^) were allocated into a subgroup containing the adipose trait after MRI, which would have been defined as a normal weight with a BMI < 25 kg/m^2^ based on the WHO criteria. On the other hand, 89 subjects (33.9% of all participants with a BMI ≥ 25 kg/m^2^) would have been classified as overweight/obese according to the WHO, and were not allocated to any adipose phenotype subgroup after MRI, demonstrating a TAT equal/lower than the sex-specific median.

Subjects which were allocated to any subgroup containing the adipose component by MRI concordantly showed significantly higher anthropometric and BIA-based measures of overweight/obesity (body weight, BMI, and waist circumference observed with anthropometry, total body fat content with BIA; Table 2) compared to subjects of the other subgroups. Among the four subgroups containing the adipose phenotype observed by MRI, VAT, SAT, and TAT were highest in subjects from the sarcopenic adipose group (Table 3).

### 3.2. Sarcopenic Phenotype

In total, 179 subjects (49.3%, 57.5% male gender) demonstrated an SMFF greater than the sex-specific median and thus were classified into one of the sarcopenic phenotype subgroups (Figure 2). Subjects with an OSA phenotype showed the highest levels of myosteatosis. When coinciding with adiposity in sarcopenic adiposity or OSA, muscle mass measures by MRI were significantly higher compared to a sarcopenic phenotype without comorbid adiposity (Table 2 and Figure 4).

Regarding correlations of muscle mass estimations by MRI and BIA, the lowest measurement values of lean body and appendicular muscle mass indices by BIA were shown for subjects with a ”*normal*” phenotype and subjects with an *isolated osteopenia* and *sarcopenia* as well as osteopenic sarcopenia phenotype. In accordance with the MR-based measurements of muscle mass, BIA-based measures of lean body and appendicular muscle mass indices were significantly higher in subjects from the adiposity-containing subgroups.

According to the morphological criteria of the European Working Group on Sarcopenia in Older People (EWGSOP) [29], *n* = 158 (43.5%) subjects (101 males and 57 females) would have been defined as moderately sarcopenic, with a skeletal muscle mass normalized to body height squared (SMI = skeletal muscle mass index in kg/m^2^) of 8.51–10.75 kg/m^2^ (men) or 5.76–6.75 kg/m^2^ (women), and *n* = 119 (32.8%) subjects as severely sarcopenic, with an SMI ≤ 8.50 kg/m^2^ (men) or ≤5.75 kg/m^2^ (women). Thereby, *n* = 108 (52.6% of all non-sarcopenic participants according to EWGSOP) subjects were allocated into a subgroup containing the sarcopenic trait by MRI who would not have been defined as sarcopenic based on the EWGSOP. On the other hand, *n* = 87 (55.0% of all sarcopenic participants according to EWGSOP) subjects would have been diagnosed with sarcopenia according to the EWGSOP criteria, subjects who were not allocated to any sarcopenic phenotype subgroup by MRI.

### 3.3. Osteopenic Phenotype

Regarding the four different subgroups containing the osteopenic phenotype (in total *n =* 183 subjects (50.4%), 57.4% male gender) (Figure 2 and Figure 4), subjects with an *OSA* and *osteopenic sarcopenia* phenotype showed significantly higher amounts of BMFF as compared to subjects with an isolated osteopenic phenotype (62.6 ± 4.9% vs. 60.9 ± 4.2%, *p* < 0.005). BMFF was shown to be significantly higher when the osteopenic phenotype coincided with a sarcopenic and/or adipose trait (Table 3).

Subjects from the isolated osteopenia subgroup showed the lowest body weight, BMI, and muscle mass by MRI/BIA, reflecting the well-known frailty in these individuals.

### 3.4. Correlations

Subjects from any of the adiposity-containing subgroups showed the highest values of fasting glucose and HbA1c (Table 4 and Figure 5); subjects from the *“normal”* subgroup without any trait of the OSA complex showed the lowest values. The highest prevalence of an impaired glucose tolerance was found in the *OSA* subgroup (in total 63.0%). The highest blood pressure values were found in the isolated or comorbid adipose phenotypes and the lowest values in the *“normal”* phenotype subgroup (in the *OSA* group: 125.8 ± 15.1 mmHg vs. in the *“normal”* subgroup: 113.5 ± 14.5 mmHg). The highest values of LDL and triglycerides were found in subjects with an isolated adipose and osteopenic adipose phenotype and the lowest in subjects with a *“normal”* and isolated sarcopenic phenotype. HFF was highest in the adiposity-containing subgroups (highest in isolated adiposity: 14.0 ± 8.3%) and lowest in the “*normal”* phenotype group (4.3 ± 5.6%). *n* = 183 (50.4%) subjects were classified with FLD, with an HFF > 5.6%. The largest percentage of subjects with an FLD was found in the osteopenic adiposity (85.7%), isolated adiposity (78.6%), and osteosarcopenic adiposity (74.4%) subgroups (Table 3).

## 4. Discussion

OSA has been described as the hazardous triad of osteopenia/osteoporosis, sarcopenia, and adiposity [9]. The objective of this study was to analyze the feasibility of MR imaging biomarkers of bone, muscle, and fat in order to depict the different conditions of OSA for body composition phenotyping and cardio-metabolic risk stratification. We determined the three imaging biomarkers and used a sex-specific, median-divided approach to allocate subjects to either one phenotypic subgroup according to the OSA complex or to a “*normal*”/healthy phenotype. With regard to adiposity and sarcopenia, subgroups were compared to the well-established criteria from the WHO and EWGSOP, respectively. Subgroups were analyzed regarding the further characteristics of body composition as well as cardio-metabolic risk factors.

To date, there are no approved criteria for the definition or clinical diagnosis of OSA. Rather, there is a multiplicity of diagnostic tools and criteria as well as proposed cut-off values for the classification of the three individual components, which in general require different examinations [2,30]. In adition to mere obesity, fat redistribution with ectopic deposition into bone and muscle tissue have been described as key components of OSA. Therefore, we used BMFF and SMFF as imaging biomarkers for osteopenia/osteoporosis and sarcopenia. When comparing the sex-specific and median-divided approach used in this population-based cohort with the established WHO and EWGSOP definitions of obesity and sarcopenia [28,29], in total, 95 subjects (26.2%) (obesity) and 195 subjects (53.7%) (sarcopenia) were allocated to one phenotypic subgroup by MRI and WHO/EWGSOP criteria, respectively, which they would not have been allocated to according to the other criteria.

Regarding adiposity, differing allocations were predominantly found for participants with a BMI ≥ 25 kg/m^2^ (overweight/obese according to the WHO), which were indeed not allocated to any adipose-containing subgroup observed by MRI as demonstrating a TAT equal to/lower than the sex-specific median. In this regard, it is well-known that an elevated BMI does not necessarily equate to an increased body adiposity. This is due to the fact that BMI, as a simple weight-to-height ratio, is not able to distinguish between fat and lean body mass [31]. Also, no distinction between visceral and appendicular fat mass is possible by BMI, which is another important limitation in terms of the distinctly differing metabolic activity of those adipose compartments [32].

Regarding sarcopenia, differing group allocations were equally found for sarcopenic and non-sarcopenic participants according to the EWGSOP criteria. The low agreement may be attributable to the differing approaches used for group allocations. The EWGSOP criteria are based on BIA-derived measurements of muscle mass; the criteria used in this study were based on muscle fat content. Subjects with a high muscle fat content (allocated to sarcopenic subgroups) did not necessarily also show a lower muscle CSA [33]. This is due to the fact that with the progressive fat infiltration of muscle tissue (loss of muscle quality), mere CSA increases; however, functional capacity and strength decreases. In this context, it is important to note that both morphological criteria have the limitation of not considering muscle function/strength, which however is one of the main determinants of mobility and functionality. Whether decreasing muscle mass or increasing fatty infiltration is the more important qualitative determinant of functional deterioration still needs to be further analyzed [34].

Reported prevalence rates of OSA range broadly between 0.8 and 40% [35,36]. In addition to different cohorts, different diagnostic criteria and diagnostic tools seem to be the main causative factors for these wide ranges. In our cohort, the OSA phenotype accounted for 22.3% of the whole sample. However, since we did not apply any cut-off values but rather divided the subgroups according to the median in the respective phenotypic trait, the data cannot be compared.

Adipose individuals are known to have a greater overall body mass, also including muscle mass. These findings were confirmed, showing that subjects from any adiposity-containing subgroup had higher muscle masses by MRI as well as lean body and appendicular muscle mass indices by BIA. Focusing on sarcopenic adiposity in particular, a recent study found that subjects with sarcopenic adiposity had the highest percentage of body fat (up to 43.4%) [35]. Our results confirm these findings, showing the highest percentage of body fat by BIA for subjects with a sarcopenic adipose phenotype (36.4 ± 5.9%).

OSA has been described as an age-related disorder. In agreement with this, subjects with an OSA phenotype in this study were significantly older compared to the other subgroups. According to some authors, postmenopausal females have the highest risk for developing OSA due to their estrogen depletion [37]. This hypothesis was also confirmed by our results, showing that subjects from the OSA subgroup were not only more likely older, but also more likely female compared to the other subgroups.

Previous studies suggested that comorbid derangements of body composition, as in any combination of two or even three components of the OSA complex, may be associated with cardio-metabolic risk [38,39]. Our study confirms these results, showing that subjects with two traits of the OSA complex or the full triad featured the worst glycemic and lipid profile. However, further studies are warranted to analyze in particular differences in osteosarcopenic visceral adiposity and osteosarcopenic subcutaneous adiposity and their associations with DM.

Recent studies were engaged with the effects of physical activity with regard to body composition during aging, demonstrating beneficial effects. Especially regarding OSA syndrome, any form of physical activity (even if irregular) has been shown to improve the maintenance of BMD, muscle strength and quality, and to reduce obesity. In our study, subjects with an OSA phenotype demonstrated the lowest physical activity levels. The lack of physical activity of those with OSA may aggravate functional impairment, with a consecutively increased risk for falls and fractures, further cardio-metabolic dysregulations, loss of quality of life, and increased overall mortality.

### 4.1. Limitations

First, in this exploratory analysis, we exclusively used MR imaging biomarkers for body composition analyses and phenotyping. We were unable to consider functional measures of muscle performance (e.g., hand gripping strength) or histopathological samples (for muscle fat quantification) as these measurements have not been collected for the KORA MRI study. However, we did not aim to clinically diagnose sarcopenia but rather aimed to analyze phenotypic traits and correlated characteristics according to the MRI-derived biomarkers. Also, the comparison to BIA-based measures of muscle mass as described above showed a good agreement of the subgroups using the sex-specific, median-divided approach [33]. Furthermore, MRI is not confounded by obesity to the same extent as it is for BIA, which is specifically relevant for determining muscle mass in the setting of comorbid obesity. Second, measures of BMFF were not compared to DEXA, which is considered the current reference standard in the measurement of BMD for the definition of osteopenia/osteoporosis. Although some studies demonstrated that BMFF correlated inversely with BMD [14] and may thus be used as an imaging biomarker, other studies found conflicting results, emphasizing the need for further research [15]. Third, this study is an exploratory analysis, and, as such, the findings should be interpreted with caution. Further confirmatory studies are needed to validate these preliminary results and to establish generalizability.

### 4.2. Conclusions

The clinical significance of OSA syndrome as a comorbidity of three different disease entities has been increasingly recognized in recent years. Thereby, OSA has been shown to present with a higher overall health risk compared to the sum of its individual component traits. Ectopic adipogenesis and fat redistribution into the viscera, bone, and muscle tissue have been described as the key components in the co-development and progression of OSA. Recognizing any of the traits of the OSA complex may help to guide further diagnostics and induce early holistic and effective intervention and treatments, which might improve health outcomes. MRI-based biomarkers of bone, muscle, and fat may be especially feasible for body composition phenotyping and may therefore allow for targeted risk stratification and further cardio-metabolic risk assessment in suspected cases of OSA syndrome.

## Figures and Tables

**Figure 1 geriatrics-09-00150-f001:**
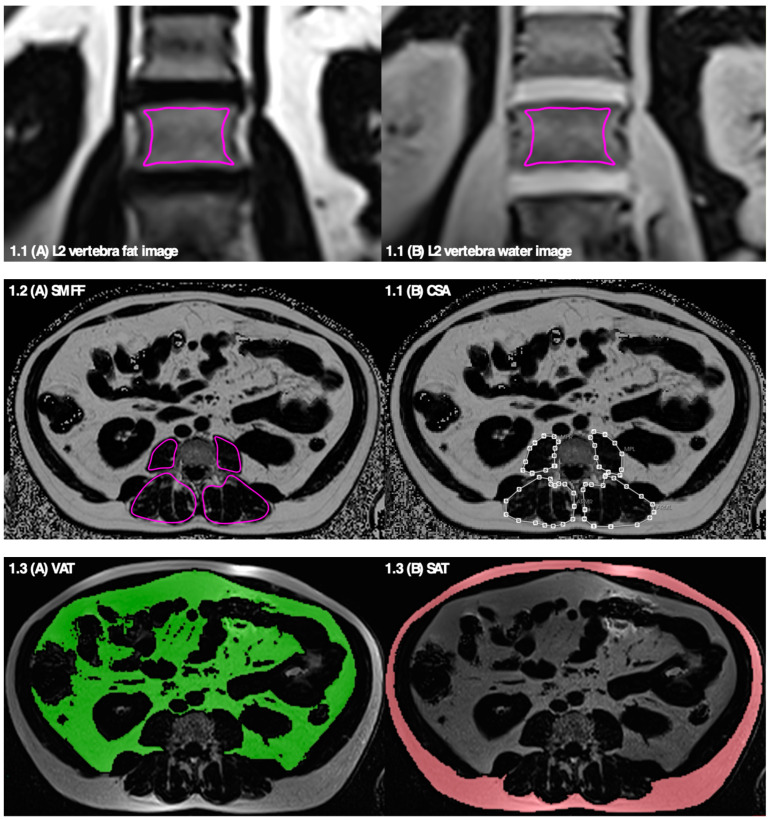
Measurements of MR imaging biomarkers of bone, muscle and fat. 1.1 Assessment of BMFF in L2 vertebra in coronally acquired 2-point DIXON Vibe sequence. ROIs were manually drawn in fat-selective images (**A**) and copied to water-selective images (**B**). 1.2 Assessment of SMFF (**A**) and CSA (**B**) of psoas muscle and autochthonous back muscles on transverse multi-echo DIXON sequence. 1.3 Assessment of VAT (**A**) and SAT (**B**) on axial dual-echo DIXON sequence.

**Figure 2 geriatrics-09-00150-f002:**
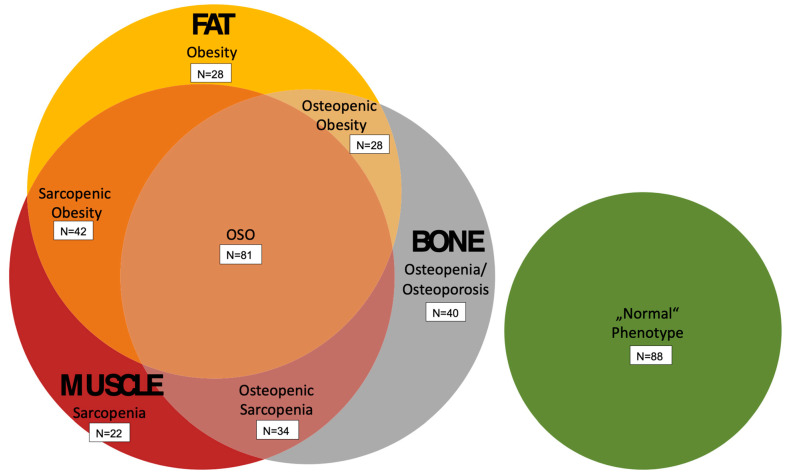
Osteopenia, sarcopenia, and adiposity according to MR imaging biomarkers of bone, muscle, and fat in 363 subjects from general population. Definitions: osteopenia with BMFF > sex-specific median (in total *n* = 183), adiposity with TAT > sex-specific median (in total *n* = 179), and sarcopenia with SMFF > sex-specific median (in total *n* = 179).

**Figure 3 geriatrics-09-00150-f003:**
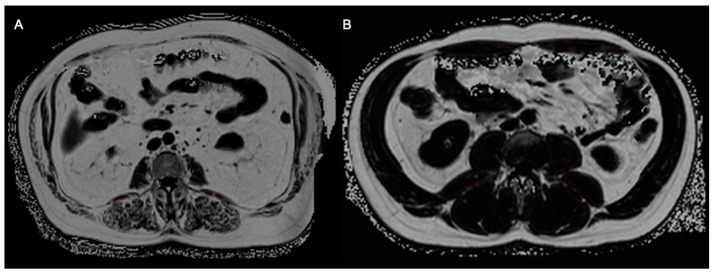
Example of two subjects from KORA study with higher (**A**) and lower (**B**) TAT, SMFF, and BMFF.

**Figure 4 geriatrics-09-00150-f004:**
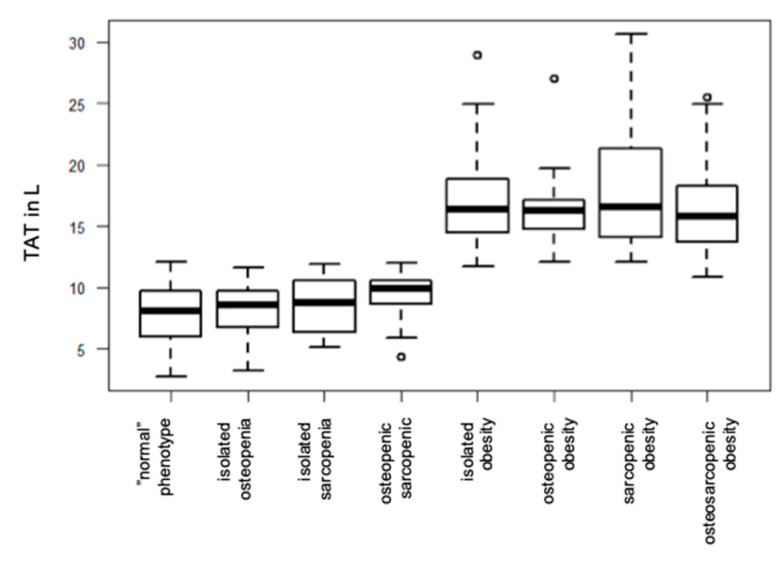
TAT, SMFF, and BMFF by MRI in study cohort.

**Figure 5 geriatrics-09-00150-f005:**
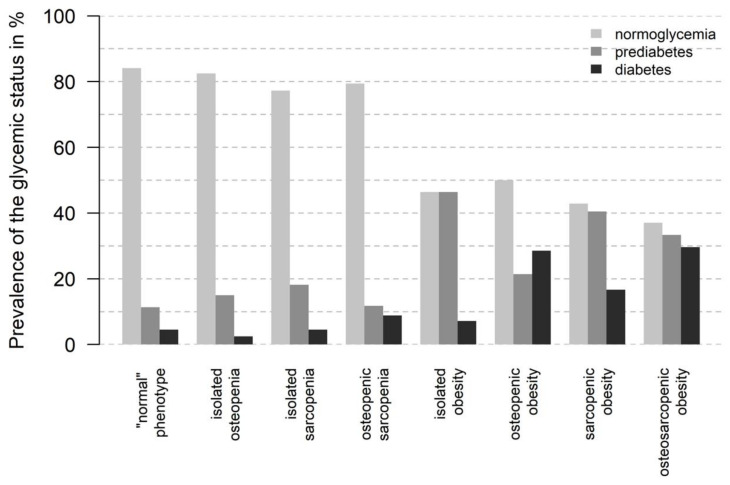
Prevalence of type 2 diabetes mellitus, prediabetes, and normoglycemia based on OGTT in study cohort.

**Table 1 geriatrics-09-00150-t001:** MR phenotypes according to OSA complex.

	**BMFF**	**SMFF**	**TAT**
sex-specific median: women men	55.8% 55.4%	13.2% 10.4%	10.8 L 12.9 L
“*Normal*” phenotype	≤sex-specific median	≤sex-specific median	≤sex-specific median
isolated osteopenia	>sex-specific median	≤sex-specific median	≤sex-specific median
isolated sarcopenia	≤sex-specific median	>sex-specific median	≤sex-specific median
osteopenic sarcopenia	>sex-specific median	>sex-specific median	≤sex-specific median
isolated adiposity	≤sex-specific median	≤sex-specific median	>sex-specific median
osteopenic adiposity	>sex-specific median	≤sex-specific median	>sex-specific median
sarcopenic adiposity	≤sex-specific median	>sex-specific median	>sex-specific median
osteosarcopenic adiposity	>sex-specific median	>sex-specific median	>sex-specific median

**Table 2 geriatrics-09-00150-t002:** Demographics and body composition by anthropometry and BIA. Overweight and obesity as defined by WHO is BMI ≥ 25 kg/m^2^ and ≥ 30 kg/m^2^, respectively. Data are presented as mean and standard deviation for continuous variables and counts and percentages for categorical variables. *p*-values are from *t*-Test.

	Whole Sample	“*Normal*” Phenotype	Isolated Osteopenia	Isolated Sarcopenia	Osteopenic Sarcopenia	Isolated Adiposity	Osteopenic Adiposity	Sarcopenic Adiposity	Osteosarcopenic Adiposity	
	*n* = 363	*n* = 88 (24.2%)	*n* = 40 (11.0%)	*n* = 22 (6.1%)	*n* = 34 (9.4%)	*n* = 28 (7.7%)	*n* = 28 (7.7%)	*n* = 42 (11.6%)	*n* = 81 (22.3%)	*p*-Value
Age in Years	56.0 ± 9.1	49.0 ± 7.3	55.3 ± 6.8	60.0 ± 8.9	62.2 ± 7.0	49.5 ± 6.0	55.6 ± 9.0	56.7 ± 7.7	62.4 ± 6.9	<0.005
Male Gender	209 (57.6%)	45 (51.1%)	23 (57.5%)	13 (59.1%)	23 (67.6%)	18 (64.3%)	20 (71.4%)	28 (66.7%)	39 (48.1%)	0.19
Female Gender	154 (42.4%)	43 (48.9%)	17 (42.5%)	9 (40.9%)	11 (32.4%)	10 (35.7%)	8 (28.6%)	14 (33.3%)	42 (51.9%)	
** *Body Composition by Anthropometry* **										
Body Weight in kg	82.4 ± 15.8	74.7 ± 11.8	71.6 ± 12.6	76.0 ± 13.8	72.2 ± 10.7	96.7 ± 14.4	92.5 ± 12.2	97.2 ± 15.3	86.0 ± 12.1	<0.005
BMI in kg/m^2^	27.9 ± 4.6	24.8 ± 2.7	24.1 ± 2.2	25.8 ± 3.3	24.8 ± 2.5	31.6 ± 3.4	30.4 ± 2.9	32.4 ± 4.9	30.4 ± 3.7	<0.005
WHO Overweight: BMI 25–30 kg/m^2^	156 (43.0%)	40 (45.5%)	15 (37.5%)	13 (59.0%)	17 (50.0%)	9 (32.1%)	12 (42.9%)	15 (35.7%)	35 (43.2%)	<0.005
WHO Obesity: BMI ≥ 30 kg/m^2^	106 (29.2%)	2 (2.3%)	0 (0%)	2 (9.1%)	0 (0%)	19 (67.9%)	16 (57.1%)	26 (61.9%)	41 (50.6%)	<0.005
Waist Circumference in cm	98.0 ± 13.7	87.3 ± 9.1	87.4 ± 9.8	92.6 ± 10.8	91.2 ± 9.4	109.1 ± 11.2	107.5 ± 7.2	111.1 ± 10.2	105.1 ± 10.2	<0.005
** *Body Composition by BIA* **										
Total Body Fat in %	32.0 ± 6.6	28.4 ± 5.5	27.9 ± 5.1	29.7 ± 4.0	28.1 ± 5.4	35.0 ± 5.5	33.7 ± 5.3	36.4 ± 5.9	36.4 ± 5.8	<0.005
Lean Body Mass Index in kg/m^2^	18.8 ± 2.5	17.7 ± 2.1	17.4 ± 2.1	18.1 ± 2.6	17.9 ± 2.2	20.5 ± 2.1	20.1 ± 1.8	20.4 ± 2.5	19.2 ± 2.4	<0.005
Appendicular Muscle Mass Index in kg/m^2^	7.8 ± 1.2	7.4 ± 1.1	7.2 ± 1.1	7.5 ± 1.3	7.4 ± 1.1	8.7 ± 1.1	8.5 ± 0.9	8.6 ± 1.2	8.0 ± 1.2	<0.005
EWGSOP Moderate Sarcopenia: SMI 8.51–10.75 kg/m^2^ (men) or 5.76–6.75 kg/m^2^ (women)	158 (43.5%)	36 (40.9%)	20 (50.0%)	10 (45.5%)	10 (29.4%)	26 (57.1%)	15 (53.6%)	19 (45.2%)	32 (39.5%)	0.373
EWGSOP Severe Sarcopenia: SMI ≤8.50 kg/m^2^ (men) or ≤5.75 kg/m^2^ (women)	119 (32.8%)	34 (38.6%)	20 (50%)	10 (45.5%)	22 (67.4%)	2 (7.1%)	6 (21.4%)	10 (23.6%)	15 (18.5%)	<0.005

**Table 3 geriatrics-09-00150-t003:** Body composition and measures of bone marrow, adipose tissue, and muscle by MRI.

	Whole Sample	“*Normal*” Phenotype	Isolated Osteopenia	Isolated Sarcopenia	Osteopenic Sarcopenia	Isolated Adiposity	Osteopenic Adiposity	Sarcopenic Adiposity	Osteosarcopenic Adiposity	
	*n* = 363	*n* = 88 (24.2%)	*n* = 40 (11.0%)	*n* = 22 (6.1%)	*n* = 34 (9.4%)	*n* = 28 (7.7%)	*n* = 28 (7.7%)	*n* = 42 (11.6%)	*n* = 81 (22.3%)	*p*-Value
** *Body Composition by MRI* ** ** ***1.** Bone Marrow* **										
Bone Marrow Fat Fraction in %	54.3 ± 10.2	43.2 ± 8.5	60.9 ± 4.2	49.5 ± 4.0	63.7 ± 5.6	49.5 ± 4.2	61.4 ± 4.5	48.8 ± 6.3	62.6 ± 4.9	<0.005
** *Body Composition by MRI* ** ** ***2.** Adipose Tissue* **										
TAT in Liters	12.5 ± 5.3	7.8 ± 2.1	8.2 ± 2.1	8.7 ± 2.2	9.4 ± 1.8	17.0 ± 3.8	16.3 ± 2.9	18.2 ± 5.0	16.2 ± 3.2	<0.005
VAT in Liters	4.5 ± 2.7	2.2 ± 1.4	2.9 ± 1.6	3.0 ± 1.6	3.6 ± 1.5	5.9 ± 2.5	6.6 ± 1.7	6.6 ± 2.7	6.2 ± 2.4	<0.005
SAT in Liters	8.0 ± 3.5	5.7 ± 1.6	5.4 ± 1.2	5.7 ± 1.2	5.8 ± 1.3	11.1 ± 3.2	9.7 ± 2.6	11.6 ± 3.8	10.0 ± 2.9	<0.005
Hepatic Fat Fraction in %	8.7 ± 7.9	4.3 ± 5.6	4.8 ± 3.1	5.1 ± 4.6	5.5 ± 4.3	14.0 ± 8.3	13.4 ± 9.1	10.5 ± 7.4	13.4 ± 8.8	<0.005
Hepatic Fat Fraction > 5.6%, Total and in % of Corresponding SubGroup	183 (50.4%)	17 (19.3%)	12 (30.0%)	7 (31.8%)	12 (35.3%)	22 (78.6%)	24 (85.7%)	30 (71.4%)	59 (74.7%)	<0.005
** *Body Composition by MRI* ** ** ***3.** Muscle* **										
Fat Fraction Psoas Muscle in %	6.6 [5.1, 8.9]	5.0 [4.2, 5.7]	5.3 [4.5, 6.6]	7.7 [6.1, 8.6]	8.4 [7.3, 10.0]	5.7 [4.9, 6.3]	6.0 [4.9, 7.2]	9.0 [7.5, 10.8]	9.3 [7.4, 11.2]	<0.005
Fat Fraction Autochthonous Back Muscles in %	16.2 [11.3, 21.5]	10.5 [8.5, 13.1]	12.1 [9.2, 14.8]	19.8 [17.1, 23.1]	21.5 [16.5, 24.8]	11.8 [9.7, 14.1]	11.1 [9.6, 13.3]	21.0 [18.4, 26.6]	22.4 [19.2, 28.3]	<0.005
Cross-sectional Area Psoas Muscle in mm^2^	1667.4 [1293.4, 2142.8]	1720.1 [1391.3, 2281.9]	1572.9 [1268.3, 2251.0]	1714.5 [1412.3, 2350.4]	1697.2 [1209.1, 2061.8]	1813.3 [1543.1, 2136.6]	1623.5 [1251.9, 2081.1]	1619.9 [1260.7, 1987.2]	1562.8 [1263.6, 1951.4]	0.13
Cross-sectional Area Autochthonous Back Muscles in mm^2^	4914.1 [4153.6, 5688.3]	4812.6 [4155.1, 5609.2]	4609.2 [3948.3, 5333.8]	5249.9 [4184.5, 5625.4]	4489.4 [3934.7, 5105.4]	5611.7 [4508.2, 6337.5]	5534.0 [5044.2, 6160.1]	5404.9 [4645.3, 6474.3]	4539.5 [3972.8, 5387.5]	<0.005

**Table 4 geriatrics-09-00150-t004:** Cardio-metabolic and musculo-skeletal characteristics of study population.

	Whole Sample	“*Normal*” Phenotype	Isolated Osteopenia	Isolated Sarcopenia	Osteopenic Sarcopenia	Isolated Adiposity	Osteopenic Adiposity	Sarcopenic Adiposity	Osteosarcopenic Adiposity	
	*n* = 363	*n* = 88 (24.2%)	*n* = 40 (11.0%)	*n* = 22 (6.1%)	*n* = 34 (9.4%)	*n* = 28 (7.7%)	*n* = 28 (7.7%)	*n* = 42 (11.6%)	*n* = 81 (22.3%)	*p*-Value
** *Glycemic Status* **										
HbA1c in %	5.6 ± 0.7	5.3 ± 0.4	5.5 ± 0.4	5.4 ± 0.3	5.6 ± 0.6	5.6 ± 0.9	5.8 ± 1.0	5.7 ± 1.3	5.8 ± 0.7	<0.005
Glycemic status										<0.005
- Normoglycemia	226 (62.3%)	74 (84.1%)	33 (82.5%)	17 (77.3%)	27 (79.4%)	13 (46.4%)	14 (50.0%)	18 (42.9%)	30 (37.0%)	
- Prediabetes	87 (24.0%)	10 (11.4%)	6 (15.0%)	4 (18.2%)	4 (11.8%)	13 (46.4%)	6 (21.4%)	17 (40.5%)	27 (33.3%)	
- Diabetes	50 (13.8%)	4 (4.5%)	1 (2.5%)	1 (4.5%)	3 (8.8%)	2 (7.1%)	8 (28.6%)	7 (16.7%)	24 (29.6%)	
** *Blood Pressure (BP)* **										
Systolic BP in mmHg	120.6 ± 16.2	113.5 ± 14.5	114.7 ± 12.4	118.6 ± 14.5	121.8 ± 18.4	124.0 ± 15.6	125.9 ± 13.9	125.5 ± 19.1	125.8 ± 15.1	<0.005
Diastolic BP in mmHg	75.4 ± 9.9	72.2 ± 9.6	72.8 ± 7.4	72.5 ± 9.4	73.2 ± 8.3	80.4 ± 8.9	80.9 ± 9.1	77.7 ± 9.6	77.1 ± 10.8	<0.005
Antihypertensive Medication	86 (23.7%)	9 (10.2%)	6 (15.0%)	4 (18.2%)	8 (23.5%)	5 (17.9%)	11 (39.3%)	14 (33.3%)	29 (35.8%)	<0.005
** *Blood Lipids and Vitamin D* **										
HDL in mg/dL	61.9 ± 17.5	64.3 ± 19.6	67.5 ± 16.1	68.4 ± 17.1	67.1 ± 16.0	50.5 ± 13.3	49.2 ± 11.0	60.0 ± 15.6	62.0 ± 17.1	<0.005
LDL in mg/dL	139.3 ± 32.4	130.4 ± 29.4	140.9 ± 36.1	135.3 ± 29.6	146.8 ± 30.2	145.8 ± 29.1	148.4 ± 34.4	134.8 ± 32.4	143.1 ± 34.1	0.05
Triglycerides in mg/dL	131.4 ± 86.1	100.9 ± 61.7	101.5 ± 52.5	89.3 ± 30.3	118.5 ± 96.4	178.0 ± 76.7	185.2 ± 98.3	142.7 ± 82.2	155.6 ± 103.9	<0.005
Lipid-Lowering Medication	37 (10.2%)	3 (3.4%)	1 (2.5%)	0 (0.0%)	6 (17.6%)	1 (3.6%)	5 (17.9%)	6 (14.3%)	15 (18.5%)	<0.005
Vitamin D (Calciferol) in ng/mL	23.5 ± 11.3	26.4 ± 13.4	26.9 ± 9.7	22.2 ± 8.5	26.4 ± 10.4	20.2 ± 11.1	20.5 ± 11.4	20.5 ± 8.6	21.5 ± 10.5	<0.005
** *Physical Activity* **										
Physical Activity										<0.005
Regularly >2 h/week	106 (29.2%)	41 (46.6%)	11 (27.5%)	4 (18.2%)	14 (41.2%)	7 (25.0%)	3 (10.7%)	10 (23.8%)	16 (19.8%)	
Regularly 1–2 h/week	112 (30.9%)	29 (33.0%)	13 (32.5%)	7 (31.8%)	9 (26.5%)	10 (35.7%)	11 (39.3%)	12 (28.6%)	21 (25.9%)	
Sporadically <1 h/week	55 (15.2%)	10 (11.4%)	5 (12.5%)	2 (9.1%)	4 (11.8%)	7 (25.0%)	4 (14.3%)	7 (16.7%)	16 (19.8%)	
Inactive	90 (24.8%)	8 (9.1%)	11 (27.5%)	9 (40.9%)	7 (20.6%)	4 (14.3%)	10 (35.7%)	13 (31.0%)	28 (34.6%)	
** *Musculo-skeletal measurements* **										
Pfirrmann Grade > 2, Disk Protrusion, or Hernia (*n* = 351)	270 (74.4%)	56 (63.6%)	34 (85.0%)	18 (81.8%)	29 (85.3%)	20 (71.4%)	18 (64.3%)	28 (66.7%)	67 (82.7%)	<0.005
Hip Cartilage Degeneration	90 (24.8%)	20 (22.7%)	2 (5.0%)	6 (27.3%)	10 (29.4%)	5 (17.9%)	8 (28.6%)	14 (33.3%)	25 (30.9%)	0.07
Somatic Pain Symptoms	140 (38.6%)	30 (34.1%)	9 (22.5%)	9 (40.9%)	11 (32.4%)	13 (46.4%)	9 (32.1%)	18 (42.9%)	41 (50.6%)	0.09

## Data Availability

Data are contained within the article.

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
