# Peer review of "MRI-Based Phenotyping for Osteosarcopenic Adiposity in Subjects from a Population-Based Cohort"

_geriatrics, 2024, doi:10.3390/geriatrics9060150_

Round 1

Reviewer 1 Report

Comments and Suggestions for Authors

This paper explores the biomarkers of bone, muscle and fat by magnetic resonance imaging (MRI) to identify osteopenia, sarcopenia and obesity as the three components of osteosarcopenic adiposity. This is a novel idea and an interesting paper. Such phenotyping of osteosarcopenic adiposity has not been done. The problem is that MRI technology is not widely available, neither it is cheap for such phenotyping to be utilized on a wide scale. However, for research purposes, it is a great idea, and the study should be published. The following are my comments:

Please conform to the latest nomenclature of the syndrome, which is osteosarcopenic adiposity (OSA), especially in this analysis where you are not necessarily limited to overweight/obesity due to the excess of fat. The infiltrated and ectopic fat/adipocytes have a more important role, as you showed in your analysis. Therefore, the statement “Overweight and obesity is characterized…..” is also not appropriate as it mostly refers to the overt overweight (as usually, although inappropriately, defined by body mass index).

There are some other limitations which authors addressed partially, but should be elaborated on with appropriate discussion and citations.

For example, among all these measurements/analyses via MRI, there were no measurements of bone mineral density (BMD) at any skeletal site. Instead, bone marrow fat-fraction (BMFF) was used as a proxy for bone quality and subsequent identification of osteopenia/osteoporosis.

Under the “Bone Marrow Fat Fraction – Osteopenia”, the authors state: “Osteopenia and/or osteoporosis, defined as reduced bone marrow density (BMD), has previously been described as the obesity of bone”. This is a wrong statement, as osteopenia and/or osteoporosis is defined as reduced bone mineral  density (BMD), not reduced bone marrow density. It is still not clear whether the fat replaces bone cells or just fills the spaces in the matrix due to the previously lost bone. Please see references by Cliff Rosen and clarify this statement. Similar situation is with the fat infiltration into the muscle fibers and rise of sarcopenic adiposity.

Please emphasize and state that this is an exploratory analysis.

Comments on the Quality of English Language

No comments

Author Response

This paper explores the biomarkers of bone, muscle and fat by magnetic resonance imaging (MRI) to identify osteopenia, sarcopenia and obesity as the three components of osteosarcopenic adiposity. This is a novel idea and an interesting paper. Such phenotyping of osteosarcopenic adiposity has not been done. The problem is that MRI technology is not widely available, neither it is cheap for such phenotyping to be utilized on a wide scale. However, for research purposes, it is a great idea, and the study should be published. 

The following are my comments:

Comment #1: Please conform to the latest nomenclature of the syndrome, which is osteosarcopenic adiposity (OSA), especially in this analysis where you are not necessarily limited to overweight/obesity due to the excess of fat. The infiltrated and ectopic fat/adipocytes have a more important role, as you showed in your analysis. Therefore, the statement “Overweight and obesity is characterized…..” is also not appropriate as it mostly refers to the overt overweight (as usually, although inappropriately, defined by body mass index).

Answer to Comment #1: We thank the Reviewer for this comment and for raising this important point. We agree and have revised our manuscript, the tables and figures accordingly. 

There are some other limitations which authors addressed partially, but should be elaborated on with appropriate discussion and citations.

Comment #2: For example, among all these measurements/analyses via MRI, there were no measurements of bone mineral density (BMD) at any skeletal site. Instead, bone marrow fat-fraction (BMFF) was used as a proxy for bone quality and subsequent identification of osteopenia/osteoporosis. Under the “Bone Marrow Fat Fraction – Osteopenia”, the authors state: “Osteopenia and/or osteoporosis, defined as reduced bone marrow density (BMD), has previously been described as the obesity of bone”. This is a wrong statement, as osteopenia and/or osteoporosis is defined as reduced bone mineral  density (BMD), not reduced bone marrow density. It is still not clear whether the fat replaces bone cells or just fills the spaces in the matrix due to the previously lost bone. Please see references by Cliff Rosen and clarify this statement. Similar situation is with the fat infiltration into the muscle fibers and rise of sarcopenic adiposity.

Answer to Comment #2: Thank you for bringing up this important topic. All data for this analysis were derived from the KORA study. Unfortunately, no bone mineral measurements or histopathological muscle tissue samples were conducted within this study, so we do not have these data available for the present cohort. However, we agree with the Reviewer, that this is a relevant limitation of the study. Also we agree, that our sentence (as stated in the comment above) needs correction and clarification. Therefore, we have revised the pertaining sections, have added another reference and have also revised the limitations section accordingly.

Comment #3: Please emphasize and state that this is an exploratory analysis.

Answer to Comment #3: We thank the reviewer for this comment and have added this in the introduction and limitations section.

Reviewer 2 Report

Comments and Suggestions for Authors

·         You want to conduct body composition analysis using magnetic resonance imaging (MRI) to create a stratification of patients with Osteosarcopenic Obesity (OSO). This approach would be advantageous because dual-energy X-ray absorptiometry (DEXA) and bioelectrical impedance analysis (BIA) do not provide certain measurements. MRI allows for the simultaneous characterization and quantification of skeletal muscle, as well as a thorough analysis of adipose tissue compartments and bone assessment (e.g., fat infiltration).

·         I think comparing the results obtained from MRI with data from DEXA would be essential. This comparison would help validate the MRI findings and demonstrate its advantages over traditional methods. It would provide a clearer understanding of how MRI's detailed analysis of body composition, particularly in patients with Osteosarcopenic Obesity, correlates with the measurements from DEXA.

·         The meanings of VAT and SAT should be mentioned.

·         The meanings of skeletal muscle fat fraction (SMFF) and cross-sectional area (CSA) should be included before Figure 1. Including these definitions will provide clarity for readers before they view Figure 1.

·         Can muscle mass also be estimated using MR imaging, in addition to providing cross-sectional area (CSA)?

·         I suggest a different analysis. You have the opportunity to establish a ranking system that consolidates all information into a single value. You could combine the values obtained from your method: BMFF + TAT + VAT + SAT + CSA (and any other relevant measures). It’s important to note that the specific units won’t matter in this context. I would expect that this single score would correlate with other health parameters, such as glucose levels. Thus, the key idea is to show that a higher score indicates worse health. Using this single score, correlations (figures) with the whole sample (363) could be improved by using different colors for each subgroup stratification. This would enhance clarity and allow for easier comparison between the subgroups, making the visual representation more informative.

·         Table 1 is fantastic, but it could be more complete by including the median values used for selection and filtering. Adding this information would provide clearer context and enhance the overall understanding of the data presented.

·         The tables could be presented more effectively. The article and data are excellent, but they are displayed in poorly structured tables. Improving the layout and organization of the tables would enhance readability and comprehension of the information.

·         I 'm glad you liked the discussion about the limitations of BMI! It's important to recognize that BMI does not necessarily equate to increased body adiposity

·         The text may be improved in lines 390-392.

·         It is clear to me that bone marrow fat (BMFF) increases continuously with age. Given that you used imaging biomarkers of bone (BMFF), muscle (SMFF), and adipose tissue (TAT) to allocate subjects to specific phenotypes according to the OSO complex, is it possible to say that some of these factors are more influenced by age  (as BMFF) while others are influenced by nutrition?

·         An illustration depicting fat redistribution into the visceral area, bone, and muscle tissue would be beneficial for readers to grasp the essence of Osteosarcopenic Obesity (OSO) syndrome. This visual representation could effectively highlight the key changes associated with the condition.

Author Response

You want to conduct body composition analysis using magnetic resonance imaging (MRI) to create a stratification of patients with Osteosarcopenic Obesity (OSO). This approach would be advantageous because dual-energy X-ray absorptiometry (DEXA) and bioelectrical impedance analysis (BIA) do not provide certain measurements. MRI allows for the simultaneous characterization and quantification of skeletal muscle, as well as a thorough analysis of adipose tissue compartments and bone assessment (e.g., fat infiltration).

Comment #1: I think comparing the results obtained from MRI with data from DEXA would be essential. This comparison would help validate the MRI findings and demonstrate its advantages over traditional methods. It would provide a clearer understanding of how MRI's detailed analysis of body composition, particularly in patients with Osteosarcopenic Obesity, correlates with the measurements from DEXA.

Answer to Comment #1: We thank the Reviewer for this comment. We fundamentally agree with these statements. Unfortunately, no bone mineral measurements (such as DEXA) were conducted within the KORA study, so we do not have these data available for the present cohort. Since we agree, that this is a  relevant limitation of the study, we have added and revised the imitations section accordingly.

Comment #2: The meanings of VAT and SAT should be mentioned.

Answer to Comment #2: Thank you very much for bringing this to our attention. Indeed, a definition of the abbreviations VAT and SAT was missing, which we have now added in the introduction section.

Comment #3: The meanings of skeletal muscle fat fraction (SMFF) and cross-sectional area (CSA) should be included before Figure 1. Including these definitions will provide clarity for readers before they view Figure 1.

Answer to Comment #3: Thank you for bringing this up. We gladly agree here as well - the placement of Figure 1 in the text was indeed somewhat unfortunate. Therefore, we have repositioned Figure 1 so that it appears only after the explanation of the biomarkers in the methods section.

Comment #4: Can muscle mass also be estimated using MR imaging, in addition to providing cross-sectional area (CSA)?

Answer to Comment #4: Thank you for this question. Indeed, we also would have liked to analyze not only the muscle cross-sectional area but also the volume/mass. However, as we analyzed the preset MRI sequences derived from the KORA study protocol, the muscles under investigation (M. psoas major and the autochthonous back muscles) were not fully captured from their origin to their insertion, making a reliable assessment of muscle mass not feasible. However, a single level-based quantification of cross-sectional area (CSA) at level L3 vertebra as performed in this analysis has been shown to be representative for the muscle mass of the entire body.

Comment #5: I suggest a different analysis. You have the opportunity to establish a ranking system that consolidates all information into a single value. You could combine the values obtained from your method: BMFF + TAT + VAT + SAT + CSA (and any other relevant measures). It’s important to note that the specific units won’t matter in this context. I would expect that this single score would correlate with other health parameters, such as glucose levels. Thus, the key idea is to show that a higher score indicates worse health. Using this single score, correlations (figures) with the whole sample (363) could be improved by using different colors for each subgroup stratification. This would enhance clarity and allow for easier comparison between the subgroups, making the visual representation more informative.

Answer to Comment #5: We thank the reviewer for this valuable suggestion. We agree that this is an excellent idea and appreciate the opportunity to pursue it further. As the current analysis represents only an exploratory, preliminary study, we believe that a more comprehensive examination of this topic would best fit in a future manuscript. With the Reviewer’s support, we propose to incorporate this analysis into a follow-up manuscript, focusing specifically on OSA traits derived from MRI-based biomarkers and their more detailed correlations with cardio-metabolic and musculo-skeletal health and disease outcomes as well as lifestyle habits (e.g. nutrition).

Comment #6: Table 1 is fantastic, but it could be more complete by including the median values used for selection and filtering. Adding this information would provide clearer context and enhance the overall understanding of the data presented.

Answer to Comment #6: Thank you for this excellent suggestion to clarify Table 1. We have added the missing information accordingly.

Comment #7:·         The tables could be presented more effectively. The article and data are excellent, but they are displayed in poorly structured tables. Improving the layout and organization of the tables would enhance readability and comprehension of the information.

Answer to Comment #7: Unfortunately, it seems that the original layout of the tables was lost during the initial submission and formatting process. We have therefore included a PDF that shows the intended formatting. If desired, we would, of course, be happy to make further adjustments.

Comment #8: I 'm glad you liked the discussion about the limitations of BMI! It's important to recognize that BMI does not necessarily equate to increased body adiposity

Answer to Comment #8: Thank you.

Comment #9: The text may be improved in lines 390-392.

Answer to Comment #9: Unfortunately, the version provided to us does not include line numbers, so we are currently unable to determine which text passage is being referenced. If you could kindly specify the sentence in question, we would be happy to make the necessary revisions.

Comment #10: It is clear to me that bone marrow fat (BMFF) increases continuously with age. Given that you used imaging biomarkers of bone (BMFF), muscle (SMFF), and adipose tissue (TAT) to allocate subjects to specific phenotypes according to the OSO complex, is it possible to say that some of these factors are more influenced by age (as BMFF) while others are influenced by nutrition?

Answer to Comment #10: Thank you for this insightful question. We agree that bone marrow fat fraction (BMFF) is well-documented to increase with age. Our exploratory analysis did not specifically focus on the effects of age or other factors (such as nutrition) with TAT and SMFF. However, we aim to conduct follow-up analyses to this preliminary work, aiming to find (independent) risk factors for osteosarcopenic adiposity/obesity and to fully explore associations with e.g. dietary habits, physical activity levels, and metabolic health. Besides that, additional research with longitudinal data (not available from the KORA study, but perspectively available from the German National Cohort study) would be required.

Comment #11: An illustration depicting fat redistribution into the visceral area, bone, and muscle tissue would be beneficial for readers to grasp the essence of Osteosarcopenic Obesity (OSO) syndrome. This visual representation could effectively highlight the key changes associated with the condition.

Answer to Comment #11: We thank the Reviewer for this excellent suggestion and have added Figure 5 accordingly.
